# Breath Alcohol Test Results in Equine Veterinarians after Performing an Abdominal Ultrasound with Ethanol

**DOI:** 10.3390/vetsci10030222

**Published:** 2023-03-14

**Authors:** Valentina Vitale, Irene Nocera, Gaby van Galen, Micaela Sgorbini, Giuseppe Conte, Bendetta Aliboni, Denis Verwilghen

**Affiliations:** 1Sydney School of Veterinary Science, University of Sydney, Camperdown, NSW 2570, Australia; 2Department of Veterinary Sciences, University of Pisa, San Piero a Grado, 56124 Pisa, Italy; 3Faculty of Veterinary Medicine, Universidad CEU-Cardenal Herrera, CEU Universities, Alfara del Patriarca, 46115 Valencia, Spain; 4Department of Agricultural, Food, and Agro-Environmental Sciences, Via del Borghetto 80, 56124 Pisa, Italy

**Keywords:** ultrasonography, alcohol inhalation, horse, breathalyser

## Abstract

**Simple Summary:**

Inhalation of alcohol vapors from hand sanitizers or the use of alcohol-based mouthwashes produces a positive breath test for alcohol for a 5–10 min period. Abdominal ultrasonography is a diagnostic tool widely used by equine clinicians and is performed with alcohol saturation. In addition, the total examination time may vary depending on many factors, and consequently, the amount of alcohol used is also variable. The aim of this study was to describe the breath alcohol test results obtained by equine veterinarians performing abdominal ultrasounds on horses. Our hypothesis was that, after the procedure, positive results are obtained after more than 5–10 min, as described for the use of hand sanitizers or mouthwashes. A total of 36 examinations were performed by six operators. The time needed to reach a negative result ranged between 0 and 60 min. The amount of ethanol used ranged between 100 and 2500 mL. A significant difference was found for the group that used more than 1000 mL compared with the groups that used less than 1000 mL. Based on this study, equine veterinarians that attend colic emergencies can test positive at the breath alcohol test for up to 60 min, especially if they use more than 1 L of ethanol.

**Abstract:**

Transcutaneous abdominal ultrasonography using alcohol saturation is used in the diagnostic workup of a variety of conditions in horses. The duration of the examination and the amount of alcohol used in each case may vary depending on several factors. The aim of this study is to describe the breath alcohol test results obtained by veterinarians performing abdominal ultrasound on horses. Six volunteers were enrolled, after written consent, and a Standardbred mare was used for the whole study protocol. Each operator performed a total of 6 ultrasounds by pouring the ethanol solution from a jar or by spray application, for a duration of 10, 30, and 60 min. An infrared breath alcohol analyzer was used immediately after completing the ultrasonography and at 5-min intervals until a negative result was obtained. Positive results were obtained for 0–60 min after the procedure. A statistically significant difference was found between the groups that used more than 1000 mL, 300–1000 mL, and less than 300 mL of ethanol. No significant differences were observed between the type of ethanol administration and the time of exposure. Based on this study, equine vets who perform ultrasound on horses can test positive at the breath alcohol test for up to 60 min following ethanol exposure.

## 1. Introduction

Transcutaneous abdominal ultrasonography is used in the diagnostic workup of a variety of conditions in horses, such as acute abdominal pain [1], recurrent colic [2,3], fever of unknown origin [4], hepatic diseases, and acute or chronic renal failure [5]. Abdominal ultrasound is a relatively easy, non-invasive tool that provides valuable information that often cannot be obtained by other diagnostic methods [1]. Although for a thorough and detailed evaluation of the abdomen, clipping of the entire flank and ventral regions is recommended [5], ultrasound can, in many instances, be done on an unclipped horse. In both circumstances (clipped and unclipped), alcohol saturation of the area to be scanned provides good contact between skin and ultrasound probe and, as such, good quality imaging [1]. The total time of examination may vary depending on the size and compliance of the horse, the reason for performing the ultrasound, the type of protocol followed, the findings, and the experience of the operator [1,5]. A fast localized abdominal sonography of horses (F.L.A.S.H.) of approximately 10 min for the examination of patients with colic symptoms has been described by Busoni et al. [6]. However, this technique is primarily used in emergency situations, but its diagnostic accuracy is limited. A full examination may be necessary in specific diseases. In this case, the abdomen is divided into 3 general regions: (1) the flank from the level of the tuber coxae to the stifle; (2) the 5–17th intercostal spaces from the lung margins to the costochondral junctions; and (3) the ventrum from the sternum to the inguinal region (from left to right). Each region is thoroughly evaluated in a systematic fashion, and each intercostal space is evaluated from the ventral lung margin to the costochondral junction to ensure imaging of all abdominal structures and the ventral thorax to detect possible thoracic abnormalities [1]. This examination, depending on the alterations detected, can be quite extensive and last 30–60 min. Consequently, the amount of alcohol used in each specific case can also be highly variable.

Breath alcohol testing is widely used in law enforcement procedures for determining the breath alcohol concentration (BrAC) and consequently the blood alcohol concentration (BAC) [7,8]. Inhalation of alcohol vapors from hand sanitizers or the use of alcohol-based mouthwashes can produce BrAC sufficient to give positive alcohol concentration results during a breath alcohol test [9]. Nevertheless, in several studies [10,11,12], it was demonstrated that the observation of a 5–10 min deprivation period prior to the test eliminated any possibility of interference. The duration and amount of alcohol vapor exposure for a clinician performing abdominal ultrasonography in horses is likely to be far higher compared to the use of hand sanitizer or mouth washes. Thus, the procedure is likely to fabricate similar false-positive results. However, this has so far not been objectively studied. Colic is one of the most frequent emergencies necessitating veterinary attention, and abdominal ultrasound is part of the diagnostic workup [13]. Emergencies can occur at any time of the day or night, so it is possible that an equine practitioner is stopped at a police checkpoint for breath alcohol testing after attending a horse with colic and performing abdominal ultrasonography.

The aim of this study was to describe the breath alcohol test results in equine veterinarians performing abdominal ultrasounds in horses and to determine for how long positive results can remain. Our hypothesis was that a positive BrAC is obtained after the ultrasound procedure and that the duration of a positive result is prolonged compared with the 5–10 min described for the use of hand sanitizers or mouth washes. Furthermore, it will be tested whether the method of alcohol application (jar or spray), the time of the ultrasound procedure, or the amount of alcohol used can influence the duration of positive test results.

## 2. Material and Methods

### 2.1. Subjects

Six operators were prospectively enrolled between December 2021 and February 2022, after written consent and approval by the Committee on Bioethics of the University of Pisa (Review n. 11, 28 May 2021) and Research Ethical Committee (n. 29913/18). They were 4th and 5th year veterinary students (five females and one male) between 22 and 26 years old. They were all healthy, with a healthy weight range (body mass index: 18.5–24.9) and reported habitual alcohol consumption between zero and two drinks per week. They were asked not to consume alcoholic drinks in the 24 h prior to the procedure; no indications were given regarding food consumption.

### 2.2. Procedure

A 20-year-old Standardbred mare, owned by the University of Pisa, was used for the ultrasonography on different days throughout the study. Only one ultrasound per day was performed to avoid multiple exposures to alcohol vapor. The examinations were carried out in the morning between 9:00 and 12:00 AM and always inside the same examination room of 35 m^2^. The room was closed during the procedure, but ventilation was allowed immediately at the end of the ultrasound. Room temperature was approximately 21 °C during the entire study protocol, as set by the institution’s heating system. Ethanol solution (90%) was used for proper contact between the probe and the skin. Each operator performed three ultrasounds by pouring the ethanol solution from a jar and three by spray application, for the duration of 10, 30, and 60 min, leading to a total of six examinations per operator. The duration of the exam and the type of alcohol application were randomized by a coin toss. Operators wore powder-free nitrile examination gloves (MedLine Sensicare^®^) during the ultrasonography to avoid accidental dermal absorption of the alcohol solution. Furthermore, a wash-out period of at least 24 h between each ultrasound was set.

The ultrasonography was carried out as described elsewhere [1] and included the visualization of the left kidney, spleen, inguinal region, stomach, and liver on the left side and of the cecum, right kidney, inguinal region, right dorsal colon, duodenum, and liver on the right side. The amount of ethanol used for each session was recorded in mL and subsequently classified into three categories: <300 mL; 300–1000 mL and >1000 mL.

### 2.3. Breath Alcohol Test

An evidential infrared breath alcohol analyzer (Alcotest 7110, Drager, Lubeck, Germany) was used in accordance with manufacturer instructions. The device was bought with the funding of the University of Sydney (Murray Bain Horse Fund). This breathalyzer type uses an electro-chemical sensor and an infrared-optical sensor (9.5 μm wavelength), and its use is approved both in Europe and in Australia. The instrument was calibrated prior to the beginning of the study and was programmed to provide a readout of the estimated BAC (g/100 mL or %) using a blood-to-breath ratio of 2000:1. The results were grouped into three categories: (1) green or negative when alcohol concentration was less than 0.019% (GR); (2) yellow or positive but within the tolerated legal level to drive when the result was between 0.019 and 0.050% (YR); red or positive and outside the legal level to drive when breath alcohol concentration was more than 0.050% (RR).

Once the ultrasonography was completed, the horse was immediately removed from the examination room, and the door to the room was left open. The alcohol test was carried out inside the same examination room in front of the open door to allow air circulation but to avoid a possible effect of different meteorological conditions and environmental temperature on different days on test results.

Immediately after completing the ultrasonography (T0) the operators discarded the gloves used for the ultrasound and worn new single-use gloves to avoid external contamination of the device and mouthpiece. They opened a new mouthpiece, placed it on the device, and provided a first breath sample immediately. After any positive result, either YR or RR, the test was repeated at 5-min intervals (T5, T10, T15, and so on) until a GR was obtained. A single mouthpiece was used for all the tests in each day session, as condensate in the mouthpiece was demonstrated not to influence test results [9,14]. The mouthpiece was discarded at the end of the session after the first negative result was obtained, and a new mouthpiece was opened for the following session of the study protocol.

### 2.4. Statistical Analysis

The value obtained at T0, the total time needed to reach a negative result (<0.019%), the total time spent with a result above the tolerated legal level to drive (>0.050%), and the total time spent with a result within the tolerated legal level to drive (0.019–0.050%) were calculated for the groups by type of ethanol administration (jar or spray), the volume of ethanol used (<300 mL, 300–1000 mL, and >1000 mL), and the exposure time (10, 30 and 60 min).

These results were analyzed by a three-way analysis of variance, which considered the effects of the type of ethanol administration (jar or spray), the volume of ethanol used, and the exposure time, according to the following linear model:y_ijz_ = μ + A_i_ + V_j_ + E_z_ + ε_jz_(1)where:

y_ijz_ = variables;

μ = mean;

A_i_ = fixed effect of the ith type of ethanol administration (tank, spray);

V_j_ = fixed effect of the jth class of ethanol volume used (<300, 300–1000, >1000 mL);

E_z_ = fixed effect of the zth class of exposure time to ethanol (10, 30, 60 min);

ε_jz_ = random error.

Differences were declared significantly different at a *p*-value < 0.05. When the interaction effect was significant, a post hoc Tuckey’s analysis was done.

## 3. Results

A total of 36 ultrasonographic examinations were performed by the six operators. The amount of ethanol used in each session ranged from 100 up to 2500 mL, with a median value of 500 mL. At T0, there were 11/36 RR (30%), 19/36 YR (53%) and 6/36 GR (17%). The time needed to reach a negative result ranged from 0 to 60 min, with a median of 7.5 min. Positive results above the tolerated legal level to drive (>0.05%) were obtained for 0–35 min with a median of 0 min, while positive results within the tolerated legal level to drive (0.019–0.05%) were obtained for 0–55 min with a median of 5 min.

Results grouped by type of ethanol administration, amount of ethanol used, and exposure time are reported in Table 1.

A statistically significant difference was found for the results obtained at T0 between the groups that used more than 1000 mL, 300–1000 mL, and less than 300 mL of ethanol. Furthermore, the time spent with a RR was prolonged for the groups that used more than 1000 mL and 300–1000 mL compared to the group that used less than 300 mL. The time spent with a YR and the time necessary to reach a GR were longer for the group that used more than 1000 mL compared to the other two.

No significant differences were observed between the type of ethanol administration and the time of exposure.

The complete results of the statistical analysis are reported in Table 2.

## 4. Discussion

This study is the first to show that a person can test positive on a breath alcohol test after performing an abdominal ultrasound on horses. Indeed, at T0 83 % of the time, a positive result was obtained, and all participants tested positive at least 4/6 times.

Previous research that studied the effect of hand sanitizer on breath alcohol tests concluded that the positive breath test results reported by the instrument were due to alcohol contamination of the respiratory dead space rather than alcohol absorption [9,12]. This was supported by the timeframe of the response, as positive results were not reported for longer than 15 min. For this reason, law enforcement officers are trained to conduct an observation period that lasts 15 min to allow any potential mouth alcohol contamination to dissipate [9]. However, in some of our scenarios, this timeframe would not have been enough. Indeed, we obtained positive results for quite a prolonged time, in particular when more than 1 L of ethanol was used. In that case, 100 % of our participants returned positive at T0 and the median result at T30 was still 0.023 %. Unfortunately, we did not measure BAC, and thus we do not know if these positive results corresponded or not with some degree of alcohol absorption in the blood. Nevertheless, 60 min seems to be too long a time for clearing respiratory dead space; therefore, a degree of alcohol absorption through the lungs is suspected at least in some of our operators.

Exposure to alcohol vapor is often assumed to pose a negligible risk; however, many studies have reported that alcohol inhalation can result in systemic absorption and, although it may not raise the blood alcohol level as much as oral consumption, there is a quick increase in blood levels following exposure [15,16]. It is also not uncommon for clinicians to report that they feel the systemic effects of alcohol exposure while performing ultrasonography (personal experience GG).

Alcohol inhalation in people has been documented within contexts associated with incidental exposure (e.g., occupational or environmental) as well as intentional (while using devices that deliver alcohol vapor, such as asthma inhalers or *e*-cigarettes) [16]. However, to date, no studies have been conducted on the possible accidental exposure of large animal vets.

Compared with oral administration, alcohol vapor exposure provides a faster, more reliable method of inducing alcohol dependence in animal models [17]. Although the pharmacodynamics of inhaled alcohol have not been established, it is possible that inhalation of alcohol vapor may correspond to low levels of total exposure that share features of the rising curve after acute oral alcohol exposure [16]. Thus, although this was beyond the scope of this study, alcohol inhalation may constitute a consistent risk of inadvertent alcohol exposure for equine vets. Although the impairments associated with low-level alcohol exposure may not be sufficient to endanger personal safety (e.g., driving under the influence), subjective effects could trigger the well-studied alcohol-specific expectations that may increase reactivity to alcohol cues and motivate drinking behavior [16]. Furthermore, chronic intermittent alcohol vapor exposure in rats produces widespread significant tissue injury, including hepatic, pulmonary, and cardiovascular changes [18]. Some equine veterinarians, in particular those specializing in internal medicine, perform frequent abdominal ultrasounds on their patients with ethanol and are probably unaware of the possible side effects associated with the procedure.

This study presents several limitations, in particular the small sample size, which may not allow us to generalize our results to a wider population. The people included in this research protocol were all of similar age; most were female and had similar alcohol consumption habits. Results could be completely different with a different study population [19]. Secondarily, as we did not measure the BAC of the volunteers involved in the study, we can only hypothesize that some degree of lung absorption took place in some cases, but we do not have any objective measure to support that. Furthermore, the study was only performed during the morning hours (between 9:00 and 12:00 AM), and results may be different in the afternoon or evening. Alcohol in the blood is metabolized faster in the evening compared to mornings or early afternoons, based on diurnal variation in hepatic alcohol dehydrogenase activity [20]. Thus, it is possible that performing the same study in the evening hours would have led to different results. Furthermore, previous studies demonstrated that alcohol is metabolized slower on an empty stomach, even when alcohol is administered intravenously, as there is an increased activity of alcohol metabolizing enzymes and increased hepatic blood flow for over three hours after a meal [21]. The lack of information regarding the food consumption of the volunteers prior to the study procedure may be responsible for inter- and intra-individual variability [20,21,22].

Finally, in order to avoid the effects of different environmental temperatures, humidity, and ventilation, the study was designed to take place in a closed room. This room was the same where the examination took place, so it is possible that there was some influence of the residual alcohol in the air and environment, although the room was big, and the operator moved towards the open door on the opposite side where the ultrasonography took place. During a field examination of an equine patient, environmental conditions may be extremely different from those of this study, according to the geographical location, the season, and the weather; thus, the results cannot be extrapolated to other situations.

## 5. Conclusions

Based on this study, equine vets that attend colic emergencies can test positive at the breath alcohol test up to 60 min following ethanol exposure, especially if more than 1 L of ethanol was used. It is recommended to wait at least 35 min before driving after performing an abdominal ultrasound.

Further studies with a larger sample of people under field conditions are needed to confirm these results and to assess the possible alcohol vapor inhalation and its consequences to the health of veterinarians.

## Figures and Tables

**Table 1 vetsci-10-00222-t001:** Descriptive results of the statistical analysis of the type of ethanol administration, volume of ethanol used, and time of exposure, expressed as mean, median, standard deviation (SD), and minimum and maximum value.

	Mean	Median	SD	Min	Max
Spray					
T0 value (%)	0.04	0.03	0.02	0.008	0.082
T+/− (min)	14.44	5.00	17.39	0	55
T red (min)	1.38	0.00	2.30	0	5
T yellow (min)	13.05	5.00	16.37	0	55
Jair					
T0 value (%)	0.04	0.04	0.02	0.016	0.087
T+/− (min)	18.88	15.00	16.23	0	60
T red (min)	4.44	0.00	9.68	0	35
T yellow (min)	14.44	10.00	13.27	0	35
<300					
T0 value (%)	0.02	0.02	0.01	0.008	0.032
T+/− (min)	2.69	5.00	2.59	0	5
T red (min)	0.00	0.00	0.00	0	0
T yellow (min)	2.69	5.00	2.59	0	5
300–1000					
T0 value (%)	0.04	0.04	0.02	0.019	0.082
T+/− (min)	21.79	17.50	16.24	5	55
T red (min)	1.43	0.00	2.34	0	5
T yellow (min)	20.35	17.50	15.99	5	55
>1000					
T0 value (%)	0.06	0.06	0.02	0.042	0.087
T+/− (min)	28.89	30.00	15.96	5	60
T red (min)	9.44	5.00	12.10	0	35
T yellow (min)	19.44	25.00	14.46	0	35
10 min					
T0 value (%)	0.02	0.02	0.01	0.008	0.032
T+/− (min)	3.75	5.00	4.33	0	15
T red (min)	0.00	0.00	0.00	0	0
T yellow (min)	3.75	5.00	4.33	0	15
30 min					
T0 value (%)	0.04	0.04	0.02	0.016	0.082
T+/− (min)	24.17	27.50	15.93	0	55
T red (min)	1.67	0.00	2.46	0	5
T yellow (min)	22.50	25.00	15.59	0	55
60 min					
T0 value (%)	0.06	0.06	0.02	0.023	0.087
T+/− (min)	22.08	15.00	18.52	5	60
T red (min)	7.08	5.00	11.17	0	35
T yellow (min)	15.00	7.50	15.23	0	40

T0 value (%): result at T0; T+/−: time necessary to reach a negative result; T red: time spent with a result > 0.050%; T yellow: time spent with a result between 0.019% and 0.050%.

**Table 2 vetsci-10-00222-t002:** Results calculated by the three-way analysis of variance, as the effect of the type of ethanol administration, volume of ethanol used, and time of exposure, expressed as averages with the relative standard error of the mean and *p*-value.

	**Type of Ethanol Administration**			**Statistical Results**	
Times	Jar		Spray	SEM	*p*-value
T0 value (%)	0.04		0.05	0.00	0.15
T+/− (min)	17.55		19.35	3.49	0.74
T red (min)	3.65		3.24	1.51	0.86
T yellow (min)	13.90		16.11	3.25	0.66
	**Amount of Ethanol Used**			**Statistical Results**	
Times	<300 mL	300–1000 mL	>1000 mL	SEM	*p*-value
T0 value (%)	0.02 ^c^	0.04 ^b^	0.06 ^a^	0.00	0.02 *
T+/− (min)	3.65 ^b^	22.44 ^a^	29.26 ^a^	4.88	0.00 *
T red (min)	0.92 ^b^	1.74 ^b^	7.66 ^a^	2.11	0.02 *
T yellow (min)	2.72 ^b^	20.70 ^a^	21.59 ^a^	4.54	0.00 *
	**Time of Exposure**			**Statistical Results**	
Times	10 min	30 min	60 min	SEM	*p*-value
T0 value (%)	0.04	0.04	0.05	0.00	0.35
T+/− (min)	15.42	21.34	18.58	4.52	0.65
T red (min)	2.38	2.36	5.59	1.96	0.40
T yellow (min)	13.04	18.98	13.00	4.21	0.39

T0 value (%): result at T0; T+/−: time necessary to reach a negative result; T red: time spent with a result >0.050%; T yellow: time spent with a result between 0.019% and 0.050%. * indicates a significant difference between groups (*p*-value < 0.05). Different superscript letter on the same line indicates a significant difference.

## Data Availability

The data that support the findings of this study are provided in the article. Further information is available from the corresponding author upon reasonable request.

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
