# Peer review of "Breath Alcohol Test Results in Equine Veterinarians after Performing an Abdominal Ultrasound with Ethanol"

_vetsci, 2023, doi:10.3390/vetsci10030222_

Round 1
Reviewer 1 Report
This report addresses if breath alcohol test could be positive after performing an abdominal ultrasound with ethanol. The topic if highly interested in equine medicine as abdominal ultrasound is a common procedure in equine clinics.
The length and structure are correct, and in general terms, it is effortless to understand. The small sample size, as the authors explain in their limitations, make it not possible extrapolate data regarding to the gender (if females were more sensitive than males). It is not neither included data from the regarding food consumption of the volunteers, something that can bias the results. Anyway, the article is very interesting for the equine practitioners.
Author Response
Many thanks for your comments.
Reviewer 2 Report
Interesting study, well designed and presented. Thank you.
Two very minor edits suggested:
Numbers less than ten should be written out as words not numbers in the text.
Ensure all years of publication in the reference list are similarly formatted (bold)
Author Response
Many thanks for your comments.
We'll check the numbers and write them in words and also the format of the references.
Thanks
Reviewer 3 Report
This interesting study investigates for the first time the effects of vet's alcohol exposure, during equine abdominal ultrasound, on alcohol test results. This is quite a new topic that warrant further attention since it can have useful practical implications and can help to improve the awareness of the potential risks associated with this profession.
The study is well constructed and the design is appropriate; I suggest to specify not only the aim of the study but also the initial hypothesis of the study. Specific comments are reported below.
Line 70: I would replace "people" with "equine veterinarians" considering the small number of operators included
Line 79-81: Are there papers in literature reporting potential effects of age, sex, weight on alcohol test results? please comment about that. What is the "normal body condition"? Please define it or add a reference.
Line 83: Are there papers reporting potential effects of diet on alcohol test results?
Line 95: Please specify what "type of the exam" means to make it clearer to the reader; does it refer to the duration? the quantity of alcohol, the type of alcohol application? to all of them?
Lines 117-120: Please comment about the potential effect of the staying of the operator in the same room where the ultrasound was performed on the results of alcohol test
Lines 155-159: is the percentage reported referred to the operators, type of the exam, alcohol application modality, all of them? please specify. Same for the discussion (Lines 184-185)
Table 1 and Table 2 : Please describe in the table legend what T0 value(%) and T+/- (min) mean
Lines 163-164: were the results significantly higher in >1000 ml group compared to others? please specify it
Lines 226-227: please comment also about the marrow age of vets included in the study, their alcohol consumption habits, their sex, their weight and the possible influence of these factors on alcohol test results
Line 247: "Based on this study, equine vets that attend colic emergencies in the field..." Considering the study setting (a closed room) the results more likely can be relevant for vets working in a clinic compared to vet working in the field
Author Response
This interesting study investigates for the first time the effects of vet's alcohol exposure, during equine abdominal ultrasound, on alcohol test results. This is quite a new topic that warrant further attention since it can have useful practical implications and can help to improve the awareness of the potential risks associated with this profession.
Thank you for your comment.
The study is well constructed and the design is appropriate; I suggest to specify not only the aim of the study but also the initial hypothesis of the study. Specific comments are reported below.
The aim and hypothesis have been better specified in the introduction. See below the reply for each comment.
Line 70: I would replace "people" with "equine veterinarians" considering the small number of operators included
This has been changed.
Line 79-81: Are there papers in literature reporting potential effects of age, sex, weight on alcohol test results? please comment about that. What is the "normal body condition"? Please define it or add a reference.
Yes, sorry, we added it to the discussion, and we indicated some references. With “normal body condition” we intended a healthy body mass index (18.5-24.9). It has been specified now in the material and method section.
Line 83: Are there papers reporting potential effects of diet on alcohol test results?
Yes, we also added a reference in the discussion of this point.
Line 95: Please specify what "type of the exam" means to make it clearer to the reader; does it refer to the duration? the quantity of alcohol, the type of alcohol application? to all of them?
Yes, sorry, we mean duration and type of alcohol application. The amount of alcohol used was recorded and used for the statistics but it was not set at the beginning of the study. It has been specified now.
Lines 117-120: Please comment about the potential effect of the staying of the operator in the same room where the ultrasound was performed on the results of alcohol test
This has been commented in the discussion.
Lines 155-159: is the percentage reported referred to the operators, type of the exam, alcohol application modality, all of them? please specify. Same for the discussion (Lines 184-185)
The percentages reported refer to the total of the exams as the abbreviations GR, YR and RR stand for green result, yellow result and red result.
Table 1 and Table 2 : Please describe in the table legend what T0 value(%) and T+/- (min) mean
Yes, sorry, this has been cut while copying the tables in the file of the manuscript. It has been added as it was supposed to be in the legend of the table: “T0 value (%): result at T0; T+/-: time necessary to reach a negative result; T red: time spent with a result > 0.050 %; T yellow: time spent with a result between 0.019 % and 0.050 %”
Lines 163-164: were the results significantly higher in >1000 ml group compared to others? please specify it
Yes, it can be seen in table 2 and, as stated in results, “a statistically significant difference was found for the results obtained at T0 between the groups which used more than 1000 ml, 300-1000 ml and less than 300 ml of ethanol. Furthermore, the time spent with a RR was prolonged for the groups of more than 1000 ml and 300-1000 ml compared to the group which used less than 300 ml. The time spent with a YR and the time necessary to reach a GR were longer for the group which used more than 1000 ml compared to the other two.”
Lines 226-227: please comment also about the marrow age of vets included in the study, their alcohol consumption habits, their sex, their weight and the possible influence of these factors on alcohol test results
Yes, we added a comment in the discussion.
Line 247: "Based on this study, equine vets that attend colic emergencies in the field..." Considering the study setting (a closed room) the results more likely can be relevant for vets working in a clinic compared to vet working in the field
You are right, we removed the words “in the field” from the sentence.